# Plant-Based Nutritional Supplementation Attenuates LPS-Induced Low-Grade Systemic Activation

**DOI:** 10.3390/ijms22020573

**Published:** 2021-01-08

**Authors:** Jin Yu, Hong Zhu, Saeid Taheri, William Mondy, Stephen Perry, Mark S. Kindy

**Affiliations:** 1Department of Pharmaceutical Sciences, College of Pharmacy, University of South Florida, Tampa, FL 33612, USA; jinyu@usf.edu (J.Y.); hongzhu@usf.edu (H.Z.); taheris@usf.edu (S.T.); wmondy@usf.edu (W.M.); 2NutriFusion^®^ LLC, Naples, FL 34109, USA; sperry@consealint.com; 3Department of Neurology, College of Medicine, University of South Florida, Tampa, FL 33620, USA; 4James A. Haley VA Medical Center, Tampa, FL 33612, USA; 5Shriners Hospital for Children, Tampa, FL 33612, USA

**Keywords:** diet, LPS, inflammation, metabolism, immune system

## Abstract

Plant-based nutritional supplementation has been shown to attenuate and reduce mortality in the processes of both acute and chronic disorders, including diabetes, obesity, cardiovascular disease, cancer, inflammatory diseases, and neurological and neurodegenerative disorders. Low-level systemic inflammation is an important contributor to these afflictions and diets enriched in phytochemicals can slow the progression. The goal of this study was to determine the impact of lipopolysaccharide (LPS)-induced inflammation on changes in glucose and insulin tolerance, performance enhancement, levels of urinary neopterin and concentrations of neurotransmitters in the striatum in mouse models. Both acute and chronic injections of LPS (2 mg/kg or 0.33 mg/kg/day, respectively) reduced glucose and insulin tolerance and elevated neopterin levels, which are indicative of systemic inflammatory responses. In addition, there were significant decreases in striatal neurotransmitter levels (dopamine and DOPAC), while serotonin (5-HT) levels were essentially unchanged. LPS resulted in impaired execution in the incremental loading test, which was reversed in mice on a supplemental plant-based diet, improving their immune function and maintaining skeletal muscle mitochondrial activity. In conclusion, plant-based nutritional supplementation attenuated the metabolic changes elicited by LPS injections, causing systemic inflammatory activity that contributed to both systemic and neurological alterations.

## 1. Introduction

Poor eating habits and diets enriched in high cholesterol or fats have become a major health issue and represent a significant cause of disability and mortality in the United States and around the world [1,2,3]. An unhealthy diet is a major risk factor for a number of chronic diseases, including diabetes, obesity, cardiovascular disease, cancer and even links to neurological and neurodegenerative diseases where alterations in the diet can have a profound effect on outcomes [4,5].

Various model systems have been examined to determine the impact of diets on chronic disorders and the potential links to human outcomes [6,7,8]. Studies in animal models have demonstrated that diets enriched in flavonoids, phytochemicals, carotenoids, etc., can alter the processes of diseases and may provide certain links to human disorders as clinical implications [9]. Therefore, diets are one lifestyle aspect that can have an impact on both our physical and mental well-being, contributing to reductions in diabetes, obesity, cardiovascular and cerebrovascular diseases, as well as many other disorders [10,11,12].

Low-level chronic inflammation is seen in many different disease states [13,14,15]. Inflammation is an essential component of innate immunity [16]. Inflammation is a response to cellular injury and is evident by increased blood flow, capillary dilatation, leucocyte infiltration, and the production of a host of chemical mediators, which serve to instigate the purging of toxic mediators and the repair of damaged tissue [17,18]. Attenuation or reduction in inflammation is a process involving cytokines and other anti-inflammatory factors, and is a more intricate process than the switching off of pro-inflammatory markers [19,20]. Neopterin is an important biomarker of immune system activation and is beneficial in the analysis of anti-inflammatory properties of nutrition [21]. Neopterin, which is derived from tetrahydrobiopterin (BH4) is generated in macrophages and microglia when stimulated by cytokines. A number of neurodegenerative diseases are also associated with inflammation. Neuroinflammation involves the triggering of glial cell activation and the release of inflammatory cytokines such as tumor necrosis factor-α (TNF-α), as well as interleukins-1β (IL-1β) and IL-6 [22,23,24]. These cytokines function in various ways to promote neuronal cell death in disorders like Parkinson’s disease (PD) or Alzheimer’s disease (AD), and in so doing contribute to pathologenesis [25,26]. We and others have shown that, in a number of neurodegenerative diseases, in addition to neuroinflammation, there is a systemic component that exacerbates these diseases and promotes the progression to neurodegeneration [27].

GrandFusion^®^ (GF) supplements are blends of fruits and vegetables that contain significant levels of vitamins and phytonutrients that can limit the impact of aging, cerebral ischemic injury, and traumatic brain injury by altering inflammatory markers and reactive oxygen species (ROS) [28]. In addition, we have shown that GF blends can improve physical activity mediated by antioxidant enzymes and signaling pathways [29,30]. Our previous studies have demonstrated that these diets have a multitude of anti-inflammatory, antioxidant, neuroprotective and neurogenic properties [29,30].

In the current study, our goal was to determine the influence of diets rich in vegetables and fruits on the outcomes associated with acute and chronic low-grade systemic inflammation in a mouse model using LPS as a trigger. Mice were fed diets enriched in fruits and vegetables for 2 months prior to the study and then subjected to lipopolysaccharide (LPS) injections to induce inflammation. The objective of the study was to determine if the presence of these nutraceutical and phytochemicals can limit the extent of the injury following LPS. The results demonstrated that these diets were able to limit the extent of injury triggered by LPS/inflammation. Glucose and insulin tolerance, body weight, workload, neopterin levels, systemic and neurological assessments were altered in the LPS-treated animals, but these alterations were corrected in the diet supplemented animals. These data suggest that these diets can impact the changes seen in low-level chronic inflammation in order to improve outcomes.

## 2. Materials and Methods

### 2.1. Animals Male

C57BL/6 mice (10–12 weeks, 25–30 g) from Jackson Laboratory (Bar Harbor, ME, USA) were maintained in a controlled environment (23 °C ± 1.5 °C, 12-h light/dark cycle) with water and food ad libitum. All studies were approved by the Institutional Animal Care and Use Committee at the University of South Florida and the Veterans Affairs Medical Center. This study adhered to the Guidelines for the Care and Use of Laboratory Animals developed by the Office of Laboratory Animal Welfare. All efforts were made to minimize the number of animals used and their suffering. Ten mice were included per experimental group, unless otherwise stated.

### 2.2. Nutritional Supplementation

Animals were fed a normal diet or a normal diet with ∼2% supplementation of NutriFusion (NF)-216 (GrandFusion—Fruit and Veggie #1 Blend, Table 1) for 2 months [26,27,28,29]. Animals were gavaged with the supplements on a daily basis, once per day. GrandFusion supplements were prepared by NutriFusion, LLC (www.nutrifusion.com). Average food intake was 3.68 ± 0.09 g/day/mouse, and the average consumption of the diets was 0.09 ± 0.006 g/day/mouse.

### 2.3. Incremental Loading Test

The incremental loading test was initiated at 10 m/min, and increased 2 m/min every 3 min, using a treadmill grade of 9%, until the mice were tired. Exercise was accomplished using a horizontal treadmill with individual lanes. Animals were allowed to run on their own without stimulation. The time to fatigue (min) and the determined exercise workload (N·m) were assessed as indexes of exercise performance. Exercise workload was calculated using the following parameters: (body mass, in kg) × (9.81 m/s^2^) × (treadmill speed, m/s) × (time of exercise, min) × (treadmill inclination, %) [31].

### 2.4. Injections and Chronic Infusions of LPS

Mice were injected intraperitoneally (i.p.) with *Escherichia coli* (*E. coli*) LPS (lot 3129; serotype 0127:B8; Sigma) 0.33 mg/kg in a volume of 10.0 mL/kg or saline (0.9% NaCl) [32]. The treatment elicited greater levels of IL-1β in the brain of LPS-injected mice (up to 100-fold; data not shown) 4 h after administration. A second group of mice had mini-osmotic pumps (Alzet^®^ Model 2002; Alza, Palo Alto, CA, USA) implanted subcutaneously between the scapulae, connected to the peritoneal cavity by a catheter, for continuous i.p. LPS infusion (0.33 or 0.83 mg/kg/day in a flow of 0.5 μL/h for 2 weeks). For mini-osmotic pump implantation, mice were anesthetized with ketamine (80 mg/kg) and xylazine (20 mg/kg) i.p. The pumps were filled with saline or LPS [33].

### 2.5. Glucose Tolerance Test

After overnight fasting, mice were fed with glucose (2 g/kg; p.o.). Blood was drawn (tail vein) at 0, 15, 30, 60, 90 and 120 min following the glucose load. Glucose concentrations were measured using a glucose meter (Optium™Xceed, Abbott, Alameda, CA, USA). The oral glucose tolerance tests (OGTT) were executed either 6 h after the acute intraperitoneal injection of LPS or at the end of the sustained LPS intraperitoneal administration. Control mice were administered with saline.

### 2.6. Insulin Tolerance Test

Mice were fasted (6 h) and then challenged with insulin (0.5 U/kg Humalog, i.p.) and blood glucose levels were measured for the insulin tolerance test (ITT). Different groups of animals (*n* = 10) were used for the analysis of OGTT and ITT.

### 2.7. Neopterin Measurement by HPLC

Neopterin levels were assayed in the urine by high-performance liquid chromatography (HPLC). Urine samples were collected following these two criteria: (i) collect urine without any direct intervention to avoid stress; or (ii) obtain pure urine without contamination with feces or animal feed. The changes in urinary neopterin levels depend on the degree of peripheral immune system activation, which is also directly connected to the intensity of exercise, duration and training status. Samples of urine were collected at 0 and 1, 3, 7, 10 and 14 days after initiation of the study. Samples were collected by the single animal method described previously. Briefly, an individual mouse urinates on plastic wrap, outside of the cage [34]. The urine is then collected into a microcentrifuge tube, quickly spun down and frozen at −80 °C. When all samples are collected, they are centrifuged at 16,000× *g* for 10 min at 4 °C and diluted in 10 volumes (*v*/*v*) of 15 mM potassium phosphate buffer, containing 5 mM EDTA. The HPLC analysis of urinary neopterin was determined using a Supercosil LC-18-T 5 μm reverse-phase column (15 × 4.6 mm), using a flow rate of 0.7 mL/min with an elution of 85% 15 mM potassium phosphate buffer, containing 15% acetonitrile, pH 6.4. The column temperature was maintained at 35 °C. The neopterin was identified and quantified by a multi-wavelength fluorescence detector (module 2475, Waters, Milford, MA, USA) with an excitation wavelength of 355 nm and an emission wavelength of 438 nm [35]. The results were expressed as μmol (neopterin)/mol creatinine. Creatinine levels were analyzed using a colorimetric kit (ThermoFisher, Waltham, MA, USA).

### 2.8. Dopamine, DOPAC and 5-HT Measurement by HPLC

Dopamine (DA), 3,4-dihydroxyphenyilacetic acid (DOPAC) and serotonin (5-HT) were analyzed in the striatum by high-performance liquid chromatography (HPLC, Alliance e2695, Waters, Milford, USA) with electrochemical detection (Waters 2465, Waters, Milford, MA, USA) at a voltage of +400 mV. Striatal samples were isolated at the end of the vehicle infusions or after sustained LPS for 2 weeks. From each group, five mice were euthanized by decapitation into liquid nitrogen. Brains were removed and the striatum was isolated and stored at −80 °C. The striatum was sonicated and centrifuged (16,000× *g* for 10 min at 4 °C) in cold 0.1 M perchloric acid. Monoamines and metabolites in the supernatants were analyzed by HPLC, as described above. The temperature of the column was maintained at 30 °C. Twenty microliters of supernatant was assessed on a 150 × 2.0 mm, 4 μm, C18 column (Synergi Hydro, Thunder Bay, ON, Canada) with 90 mM sodium phosphate, 50 mM citric acid, 1.7 mM sodium 1- heptane-sulfonate, 50 μM ethylenediaminetetraacetic acid, 10% acetonitrile, pH 3.0, with a flow of 0.25 mL/min as the mobile phase [35]. DA, DOPAC and 5-HT concentrations in the supernatants were determined as ng/mg protein.

### 2.9. Measurement of Respiratory Chain Complex I Activity

At the termination of the sustained LPS or vehicle infusions for 2 weeks, the quadriceps muscles were isolated, assembled and homogenized in 10 volumes (*v*/*v*) of 4.4 mM potassium phosphate buffer (PBS), pH 7.4, containing 0.3 M sucrose, 5 mM MOPS, 1 mM EGTA and 0.1% BSA. The homogenized samples were centrifuged at 3000× *g* for 10 min at 4 °C. The supernatant was isolated and frozen and thawed for three cycles to allow for the rupture of the mitochondrial membranes. The rate of NADH-dependent ferricyanide reduction at 420 nm (30 °C, ε = 1 mM^−1^ × cm^−1^) was used to determine the Complex I activity [36].

### 2.10. Protein Determination

Protein levels were measured by the Lowry method, using bovine serum albumin and 96-well plates.

### 2.11. Statistical Analysis

The data are presented as the mean ± SEM (standard error of mean). We analyzed the data using one-way or two-way analyses of variance (ANOVA) and post-hoc Bonferroni or Tukey’s tests, where F was significant. We used the Student’s *t*-test for independent samples when comparing two independent groups. The exercise incremental load test data were depicted as the percentage of animals that successfully completed the task at given speeds. In this case, group differences were examined by applying log-rank (Mantel–Cox). Significance for the tests was considered at *p* < 0.05. Statistics and all graphs were analyzed using GraphPad Prism 7^®^.

## 3. Results

### 3.1. Continuous LPS Injections Reduced Glucose Tolerance

Figure 1 illustrates the blood glucose and insulin levels after a single injection of LPS. LPS elevated the glucose clearance and reduced the area under the curve (AUC), suggesting a greater nutrient requirement during the time of acute inflammation when compared to the control group (Figure 1A,B). When the animals were provided with a diet enriched in fruits and vegetables, and then subjected to LPS treatment, the glucose tolerance showed a return almost to control levels. For analysis of the insulin tolerance, 6 h after a single injection of LPS (2 mg/kg or 0.33 mg/kg), mice were injected with insulin and the glycemia index was determined. Acute injection of LPS elevated insulin sensitivity (Figure 1C,D). Again, when the animals were provided a diet enriched in fruits and vegetables, and then subjected to LPS treatment, the insulin tolerance showed a return almost to control levels.

Next, we assessed the effect of repeated intraperitoneal LPS administration (0.33 mg/kg/day; 2 weeks) on glucose tolerance (Figure 2B). In contrast to acute administration (Figure 1A), the sustained injections of LPS decreased glucose tolerance, as shown in Figure 2A. No difference was found in food intake between the groups (Figure 2A). When the animals were provided a diet enriched in fruits and vegetables, and then subjected to chronic LPS treatment, the glucose tolerance showed a return almost to control levels.

### 3.2. Acute and Sustained Infusion of LPS over a Two-Week Period on Exercise Activity in Mice

The impact of acute LPS treatment and inflammation on exercise and neopterin levels in mice was determined (Figure 3). As shown in Figure 3A, in the incremental loading test (initiated 24 h after LPS, 0.33 mg/kg, i.p.), exercise performance was reduced, as seen in both the horizontal and vertical exercise workloads (Figure 3B). LPS treatment resulted in a significant impairment in the early phases of the incremental loading test, after the challenge. Pre-treatment of the mice with the nutritional diets prevented a reduction in workload, maintaining them at control levels. Continuous infusion with LPS (0.33 mg/kg/day, i.p.; Figure 3C) decreased the body weight of the mice, specifically in the first two days of the test. Figure 3D illustrates that continuous LPS-mediated inflammatory conditions elevated the neopterin levels in the urine for three days after initiating the LPS infusion. The levels returned to baseline on the seventh day of the test. Finally, the nutritional diet maintained the urinary neopterin at control levels.

### 3.3. Continuous Infusion of LPS over a Two-Week Period Provoked Inflammation and Altered Dopamine Metabolism

In mice treated with a continuous infusion of LPS, the DA levels (Figure 4A) and levels of the metabolic breakdown product, DOPAC (Figure 4B), were significantly decreased in the striatum. The exposure of the mice to the NutriFusion diet reverted the DA and DOPAC to pretreatment levels. No significant difference was detected in the 5-HT levels (Figure 4C); nevertheless, the striatal 5-HT/DA ratio was statistically elevated by LPS (Figure 4D).

### 3.4. Nutritional Diet Mitigated the Exacerbation of Immune System Initiation and Maintained Mitochondrial Activity

The impact of the NutriFusion diets on the production of neopterin and mitochondrial complex I activity in skeletal muscle were determined in mice infused with LPS and provided with daily nutritional diets. Mice were provided with these diets for 2 months and then treated with LPS (0.83 mg/kg/day, i.p.; mini-osmotic pumps for 2 weeks; Figure 5A). Figure 5A illustrates the elevated neopterin levels in the urine in the LPS treated group, which were attenuated by nutritional prophylaxis. Additionally, the NutriFusion diet temperately elevated the neopterin levels (compared to the non-inflammatory conditions). Moreover, the addition of the diet mitigated the decrease in mitochondrial complex I activity seen in the skeletal muscle of mice exposed to LPS (Figure 5B).

## 4. Discussion

In the present study, we examined the influence of a diet enriched with vegetables and fruits on the alterations in acute and chronic inflammation. We found that that consumption of this particular diet for at least a 2-month period helped to reduce the outcomes of both acute and chronic inflammation induced by LPS.

Acute inflammation can play a protective role by responding to cellular stress signals that result from immune cell activation and the injured tissue [37,38]. Low-level chronic inflammation is typified by the elevation of specific proinflammatory mediators that are associated with many chronic disorders and are important in the pathogenesis of diabetes and obesity, as well as other systemic disorders and even neurological diseases [39]. Therefore, decreasing inflammation in the body is a pathway to not only the attenuation of disease progression, but also its possible prevention. Diets enriched in phytochemicals (which are found in fruits and vegetables) are not regularly consumed because of access, cost and the fact that their quality can give rise to systemic inflammation [39]. The food pyramid, which indicates that people need to consume 2–4 servings of fruits and 3–5 servings of vegetables per day, is difficult if not impossible because of many complications. In addition, reductions in red meat, fats, oils and sweets are recommended but the majority of people in the US do not follow these guidelines. A simple, more productive approach is needed to provide the nutrition and health benefits of phytochemicals, anti-inflammatory and antioxidant complexes.

We showed in this study that chronic inflammation compromised both glucose and insulin tolerance, which is normally seen in certain chronic metabolic diseases. In addition, the administration of LPS resulted in an increase in neopterin levels, which is a marker for immune system activation. We showed that a diet enriched in fruits and vegetables (and consequently phytochemicals) was able to reverse the process and maintain and even elevate insulin sensitivity and glucose tolerance. In an acute inflammatory state, insulin sensitivity and glucose tolerance were increased, possibly due to the inhibition of glucose synthesis in the liver in connection to altered toll-like receptor (TLR) expression in target tissues [40]. LPS-mediated effects are related to an increase in TLR4 levels that triggers the activation of nuclear factor-kB (NF-kB), a transcription factor that activates a cascade of inflammatory mediators [41]. These factors control the transcription of inflammatory mediators, such as IL-1β, IL-6, TNF-α, TNF-β, INF-α, INFβ, INF-γ [42]. Inflammation of adipose tissue is an important regulatory point in the promotion of insulin resistance and obesity [42,43]. The inhibition of the downstream signaling of the insulin receptor is a critical mechanism by which inflammatory signaling leads to insulin resistance [43,44]. Inflammation can alter insulin action and give rise to diabetes and obesity by blocking insulin receptor downstream events, impairing insulin receptor substrate 1 (IRS-1) activation and phosphatidylinositol 3-kinase-dependent (PI3K) pathways, therefore compromising insulin signaling [45].

Previous studies have shown that the brain maintains homeostasis by communication with the systemic physiological system [46,47,48]. The systemic circulation is prevented from entering the brain by the blood brain barrier (BBB) and the local environment. Changes in pH, metabolic disorders, and other factors, such as infections, systemic inflammation, organ dysfunction, can alter the BBB and impact the functioning in the brain [48].

Our data demonstrated that systemic inflammation (generated by LPS) also increased neopterin levels in the urine and resulted in altered neuronal activity by decreasing dopamine (DA) metabolism. Neuroinflammation has been extensively studied over the years and the link to systemic inflammation is beginning to be observed [49,50]. Inflammation is associated with the development and progression of neurological diseases such as Alzheimer’s disease (AD), Parkinson’s disease (PD), stroke and many other diseases. We and others have suggested that systemic chronic metabolic diseases, in turn, compromise the blood–brain barrier (BBB) and bear changes in the brain that impart a sensitivity and susceptibility to neurological and neurodegenerative changes [51]. Elevated levels of inflammatory mediators in the brain can alter neurogenesis, synaptic formation and neuronal sprouting, resulting in a decrease in cognitive ability. Studies have shown that increased inflammation can result in altered DA signaling and maintenance of dopaminergic neurons.

Inputs into the basal ganglia via the striatum are necessary for the control of voluntary movement and neurons in the striatum signal the preparatory, initiation and execution of these movements [52,53]. Diseases like Parkinson’s disease (PD), occur from the loss of dopamine producing neurons which results in movement disorders leading to difficulties in motor activities. Recent studies have suggested that there is an emerging role for nutrition in PD and this may alter dopamine levels and impact the D2/D3 dopamine receptor binding [54,55]. In addition, diet can impact the microbiome to reduce inflammatory aspects that could help limit the extent of PD in individuals [55]. Foods are natural sources of chemicals that may exert critical effects on the nervous system in humans. Some of these chemicals are the neurotransmitters (NTs) acetylcholine (ACh), the modified amino acids glutamate and γ-aminobutyric acid (GABA), and the biogenic amines dopamine, serotonin (5-HT), and histamine [56]. Of interest is the ability of flavonoids to activate the extracellular signal-regulated kinase and the Akt signaling pathways, leading to the activation of the cyclic adenosine monophosphate (cAMP) response element binding protein, a transcription factor that increases the expression of a number of neurotrophins important in LTP and long-term memory. One such neurotrophin is BDNF, which is known to be crucial in controlling synapse growth, promoting an increase in dendritic spine density, and enhancing synaptic receptor density [57].

The effects of nutrition on immune system function have been studied over the years and have been shown to impact urinary neopterin levels [58]. Neopterin is a by-product of the BH4 de novo pathway and, under inflammatory conditions, the pathways involved in its generation are altered and neopterin accumulates in the blood [59]. Thus, an increase in neopterin levels has been recognized a sensitive biomarker for immune system activation. Additionally, studies have shown that dietary supplements were significantly effective in improving the redox status by effectively increasing the reduced-glutathione (GSH) levels and other reduced thiol entities while appreciably decreasing the oxidized species [60]. The beneficial outcome of the redox status was significant in the specific groups which also experienced a significant reduction in neopterin levels. To understand how this work mechanistically, elevated neopterin levels would lead to the activation of a master transcription factor nuclear factor erythroid 2–related factor 2 (Nrf2), which is a key regulator of the cellular antioxidant response. This response will further enhance the cellular antioxidant defenses, mitochondrial activity and the anti-inflammatory environment. Increased Nrf2 expression and increased downstream antioxidant proteins are increased in the brain after neopterin treatment [61,62,63]. It is suggested that selected diets: (i) prevent the aggravated increase of neopterin in an inflammatory state and (ii) dietary intake can increase the neopterin levels as a mechanism to prevent oxidative stress and inflammation. Our studies show that neopterin is elevated in chronic inflammation and nutritional supplementation can protect against this increase.

Several studies have demonstrated that diets enriched in fruits and vegetables can have anti-inflammatory and dynamic metabolic effects [27,28,29,30]. Our earlier studies have revealed that nutritional supplementation assisted in the improvement of outcomes from neurological disorders, diminished changes in age-related deficiencies and enhanced physical consequences [26,27]. We established that diets enriched in phytochemicals limited the injury and improved behavioral outcomes in mice subjected to cerebral ischemic injury [64]. Our experiments denoted that these diets were able to diminish inflammatory mediators and oxidative damage and therefore augment neuronal improvement [27,28,29,30]. We were also able to show that dietary supplementation in aged rats validated a reversal of the age-related phenomenon of elevated inflammation and reactive oxygen species (ROS) and enhanced physical activity in older animals [29]. These data insinuate that supplementation with vegetables and fruits and their associated components are necessary and sufficient to provide protection from injury and to stimulate performance in animal models.

## 5. Conclusions

In conclusion, the strengths of this study show that prolonged use of diets augmented with vegetable and/or fruit extracts mitigated systemic inflammatory conditions when used prophylactically. This demonstrates that diets enriched in anti-inflammatory mediators and other compounds are beneficial in mouse models of inflammation and could mitigate the negative influences of immune system initiation. Therefore, this aspect of the body in its adaptability to inflammation may help to alter the process of acute and chronic inflammatory disorders and protect the body from disease. The limitations of the study are that diseases in mice are not identical to man, and the potential use of nutritional supplements is restricted to prophylaxis and not therapeutic.

## Figures and Tables

**Figure 1 ijms-22-00573-f001:**
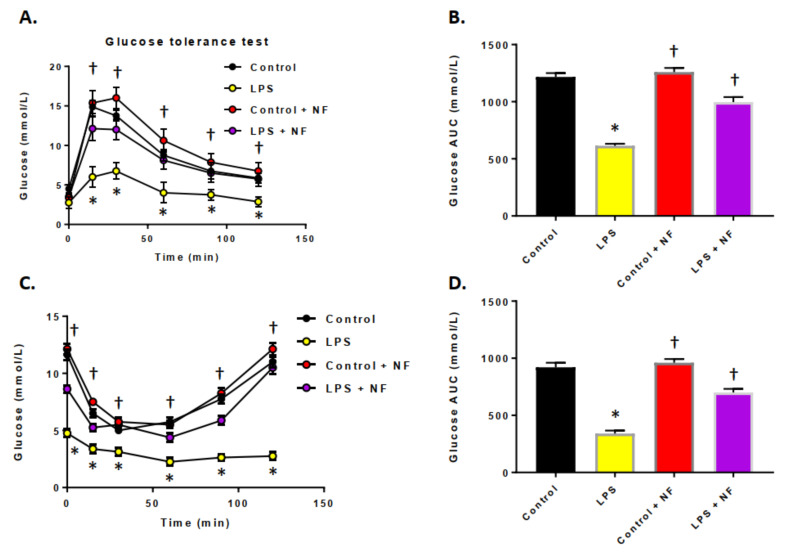
Effect of acute lipopolysaccharide (LPS) administration on glucose and insulin tolerance. (**A**) Oral glucose tolerance test (OGTT, 2 g/kg, p.o.) performed 6 h after acute intraperitoneal injection of control (saline, *n* = 10/group), LPS (2 mg/kg, *n* = 10/group, control plus NutriFusion (NF) diet (saline, *n* = 10/group), LPS + NF diet (2 mg/kg, *n* = 10/group; repeated measures one-way ANOVA followed by Bonferroni’s test (d = 1.14). (**B**) AUC of glucose response; Student’s *t*-test. (**C**) Insulin tolerance test (ITT, 0.5 U/kg, i.p.) performed 6 h after injection of control (saline, *n* = 10), LPS (2 mg/kg, *n* = 10/group), control plus NF diet (saline, *n* = 10/group), LPS + NF diet (2 mg/kg, *n* = 10/group; repeated measures one-way ANOVA followed by Bonferroni’s test (η2 = 0.108). (**D**) AUC of insulin response; one-way ANOVA followed by Bonferroni’s test. Values are mean ± SEM. * *p* < 0.01 compared to control group, † *p* < 0.01 compared to LPS group.

**Figure 2 ijms-22-00573-f002:**
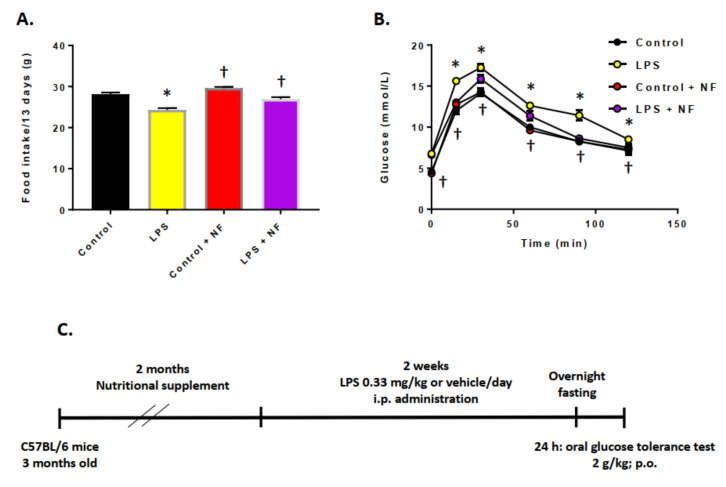
Effect of chronic lipopolysaccharide (LPS) administration on food intake and glucose tolerance. (**A**) Food intake during 13 days with repeated daily injection of control (saline, *n* = 10), LPS (0.33 mg/kg/day, i.p., *n* = 10/group), control plus NF diet (saline, *n* = 10/group), LPS + NF diet; Student’s *t*-test. (**B**) OGTT performed at the end of the repeated daily injection of control (saline, *n* = 10/group), LPS (0.33 mg/kg, i.p., *n* = 10/group), control plus NF diet (saline, *n* = 10/group), LPS + NF diet; repeated measures one-way ANOVA followed by Bonferroni’s test (*d* = − 0.62). (**C**) Diagram of experimental design; Student’s *t*-test. * *p* < 0.01 compared to control group, † *p* < 0.01 compared to LPS group.

**Figure 3 ijms-22-00573-f003:**
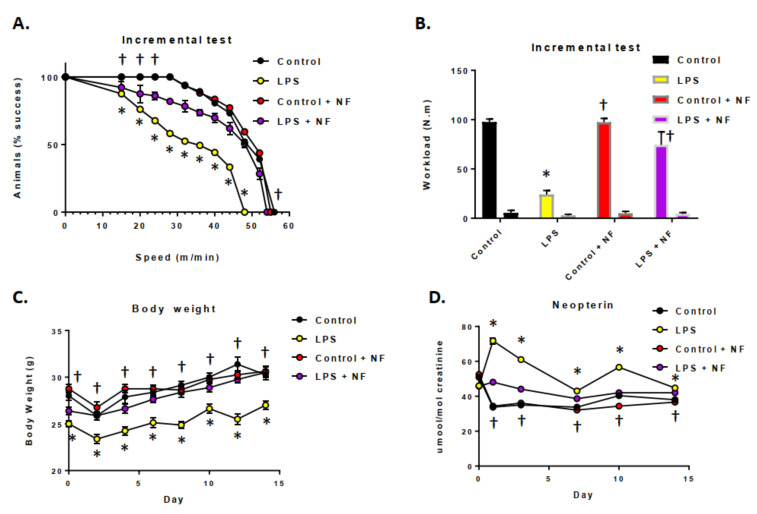
Effect of bacterial lipopolysaccharide (LPS) administration on exercise performance, and urinary neopterin levels in the striatum. (**A**) Mouse physical performance on treadmill incremental loading test (*n* = 10/group); log-rank (Mantel–Cox). The animals were adapted to the treadmill for 5 consecutive days at a speed of 10 m/min for 10 min. The incremental loading test was performed 24 h after a single dose of control (saline), LPS injection (0.33 mg/kg, i.p.), control plus NF diet (saline), LPS + NF diet (*n* = 10/group). (**B**) Vertical (lower bars) and horizontal (higher bars) workloads during incremental loading test; one-way ANOVA followed by the Bonferroni test. (**C**) Body weight and (**D**) urinary neopterin production (*n* = 10/group); repeated measures one-way ANOVA followed by Bonferroni’s test. Values are mean ± SEM. * *p* < 0.01 compared to control group, † *p* < 0.01 compared to LPS group.

**Figure 4 ijms-22-00573-f004:**
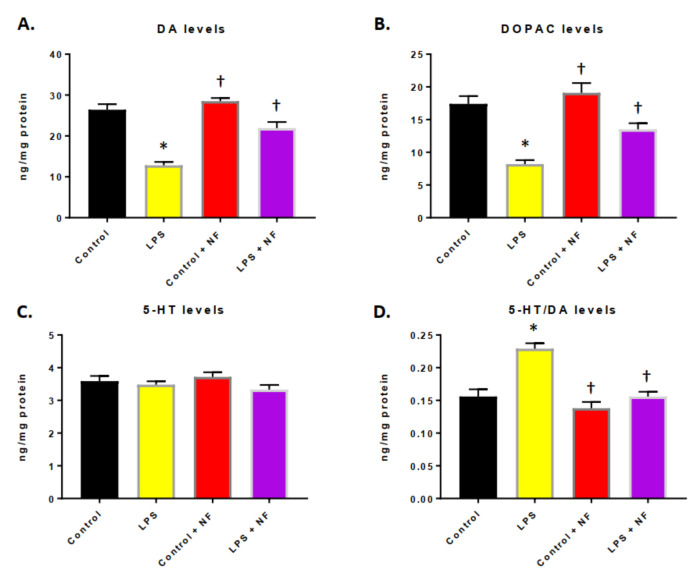
Effect of bacterial lipopolysaccharide (LPS) administration on monoamine levels in the striatum. (**A**) Dopamine (DA), (**B**) 3,4-dihydroxyphenyilacetic acid (DOPAC), (**C**) serotonin (5-HT) levels and (**D**) 5-HT/DA ratio in control (saline), LPS injection (0.33 mg/kg, i.p.), control plus NF diet (saline), LPS + NF diet (*n* = 10/group); Student’s *t*-test. Values are mean ± SEM. * *p* < 0.01 compared to control group, † *p* < 0.01 compared to LPS group.

**Figure 5 ijms-22-00573-f005:**
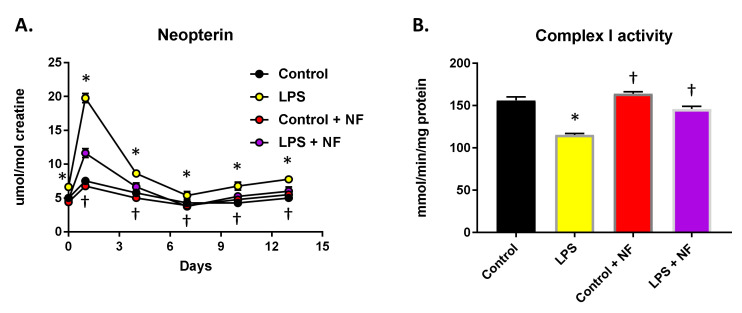
Effect of nutritional diets on inflammatory response (animals submitted to nutritional diets for 2 months). Afterwards, mini-osmotic pumps containing LPS 0.83 mg/kg/day were implanted and LPS was infused for 2 weeks. (**A**) Neopterin urinary levels; repeated measures two-way ANOVA followed by Bonferroni’s test. Groups were: control (saline), LPS injection (0.33 mg/kg, i.p.), control plus NF diet (saline), and LPS + NF diet. (**B**) Complex I activity in skeletal muscle (*n* = 10/group); two-way ANOVA followed by Tukey’s test. Values are mean ± SEM. * *p* < 0.01 compared to control group, † *p* < 0.01 compared to LPS group.

**Table 1 ijms-22-00573-t001:** Composition of GrandFusion Supplement.

t6 Essential Vitamins	% dv	Maximum Premix Claim Per 225.00 mg
Vitamin A	50.00	2500.00 IU
Vitamin C	50.00	30.00 mg
Vitamin D	50.00	200.00 IU
Vitamin E	50.00	15.00 IU
Vitamin B1	50.00	0.7500 mg
Vitamin B6	50.00	1.00 mg

Blend #1: Fruit and Vegetable Blend (NF-216), Vegetable: Powdered Tomato, Broccoli, Carrot, Shitake Mushroom, Fruit: Powdered Cranberry, Apple, Orange, Made from 100% organic materials.

## Data Availability

The data presented in this study are available on request from the corresponding author. The data are not publicly available due to privacy issues.

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
