# Peer review of "Plant-Based Nutritional Supplementation Attenuates LPS-Induced Low-Grade Systemic Activation"

_ijms, 2021, doi:10.3390/ijms22020573_

Round 1

Reviewer 1 Report

Manuscript ID: ijms-1000307:Title: Plant-based nutritional supplementation prevents excessive immune system activation

Comments

The title of the manuscript seems to indicate a review on plant-based nutrition and immune system activation.  In this study the authors reported results concerning the effect  of LPS  and NF on  glucose metabolism, neurotransmitters and mitochondrial activity,  in mice,  in a not well defined pathological setting:

  • do the authors want to evaluate the NF-supplementation effects on a model of neurological diseases? Therefore they should accordingly  modify introduction and discussion

  • do the authors want to evaluate the NF effects in a “whole body immune system activation”?  If so, evaluation of neopterin levels  is  not sufficient to the pourpose

Would the authors, according to the aims of the study,  interpret and  harmonize results in discussion, as well as  formulate  hypothesis and  suggestions for future studies?

- 2.2.nutritional supplementation:  composition of supplementation is needed, please add

- Fig.2 legend. “ Effect of acute lipopolysaccharide……..”,  is  “ acute” instead of chronic a typing mistake?  Please clarify

-Pag. 9, Fig. 4 is repeated, is it a typing mistake?

-Pag. 9: Discussion title is missing, please add it

Author Response

We thank the reviewer for the time and effort put into the review.  

Major changes:

The main goal of the study was to study acute and chronic inflammation in general as a prelude to various disorders.  We discuss in the introduction the impact of chronic low level inflammation and the impact on disease.  We tie in our other studies that show the impact of nutritional supplementation on disease.  I am not sure that we need to add or change the intro or discussion.

Minor changes:

We corrected the nutritional supplementation information.  We added a table and other information in the methods to help with the composition.

Figure 2 legend was corrected.

Figure 9 legend was corrected (should have been Figure 5 not 4).

Discussion title was miss placed in Figure 5 legend.  Corrected now.

Reviewer 2 Report

This paper has some potentially important findings but there are some gaps that should be addressed as follows: 

  1. Lack of details on the intervention Grandfusion composition and the web site had very little useful details only on vitamins.  Please provide more general information on type of fruit and vegetables, and nutrients or phytochemicals such as fiber and primary polyphenols, carotenoids and vitamins.
  2. Please provide a description of the studies strengths and weakness/limitations such usefulness of male mice as a model for human inflammatory response and other metabolic health concerns. 
  3. Please consider adding more details on the potential biological mechanisms of action based on the fruit and vegetable composition of Grandfusion

Author Response

We thank the reviewer for their input.

  1. We added detail on the nutritional supplementation (table 1 and in methods section).
  2. We added in the conclusions strengths and limitations.
  3. We do have information in the discussion that gets to the point of the reviewer as to the mechanisms.  The 3rd and 4th paragraph in the discussion talks about inflammatory mechanisms (NF-kB, cytokines, etc).